# SCALING SHARED MODEL GOVERNANCE VIA MODEL SPLITTING

## ABSTRACT

Currently the only techniques for sharing governance of a deep learning model are homomorphic encryption and secure multiparty computation. Unfortunately neither of these techniques is applicable to the training of large neural networks due to their large computational and communication overheads. As a scalable technique for shared model governance, we propose splitting deep learning model between multiple parties. This paper empirically investigates the security guarantee of this technique, which is introduced as the problem of *model completion*: Given the entire training data set or an environment simulator, and a subset of the parameters of a trained deep learning model, how much training is required to recover the model's original performance? We define a metric for evaluating the hardness of the model completion problem and study it empirically in both supervised learning on ImageNet and reinforcement learning on Atari and DeepMind Lab. Our experiments show that (1) the model completion problem is harder in reinforcement learning than in supervised learning because of the unavailability of the trained agent's trajectories, and (2) its hardness depends not primarily on the number of parameters of the missing part, but more so on their type and location. Our results suggest that model splitting might be a feasible technique for shared model governance in some settings where training is very expensive.

## 1 INTRODUCTION

With an increasing number of deep learning models being deployed in production, questions regarding data privacy and misuse are being raised (Brundage et al., 2018). The trend of training larger models on more data (LeCun et al., 2015), training models becomes increasingly expensive. Especially in a continual learning setting where models get trained over many months or years, they accrue a lot of value and are thus increasingly susceptible to theft. This prompts for technical solutions to monitor and enforce control over these models (Stoica et al., 2017). We are interested in the special case of *shared model governance*: Can two or more parties jointly train a model such that each party has to consent to every forward (inference) and backward pass (training) through the model?

Two popular methods for sharing model governance are *homomorphic encryption* (HE; Rivest et al., 1978) and *secure multi-party computation* (MPC; Yao, 1982). The major downside of both techniques is the large overhead incurred by every multiplication, both computationally, >1000x for HE (Lepoint and Naehrig, 2014; Gilad-Bachrach et al., 2016), >24x for MPC (Keller et al., 2016; Dahl, 2017), in addition to space (>1000x in case of HE) and communication (>16 bytes per 16 bit floating point multiplication in case of MPC). Unfortunately, this makes HE and MPC inapplicable to the training of large neural networks. As scalable alternative for sharing model governance with minimal overhead, we propose the method of *model splitting*: distributing a deep learning model between multiple parties such that each party holds a disjoint subset of the model's parameters.

Concretely, imagine the following scenario for sharing model governance between two parties, called Alice and Bob. Alice holds the model's first layer and Bob holds the model's remaining layers. In each training step (1) Alice does a forward pass through the first layer, (2) sends the resulting activations to Bob, (3) Bob completes the forward pass, computes the loss from the labels, and does a backward pass to the first layer, (4) sends the resulting gradients to Alice, and (5) Alice finishes the backward pass.

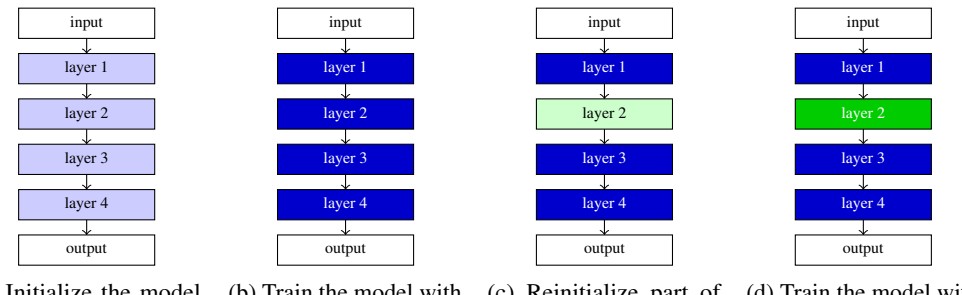

(a) Initialize the model.

(b) Train the model with training procedure $T$.

(c) Reinitialize part of the model (green).

(d) Train the model with training procedure $T'$.

Figure 1: Schematic illustration of the problem of model completion: Find the fastest retraining procedure $T'$ that recovers a loss that is at least as good as the loss from the original model (b).

How much security would Alice and Bob enjoy in this setting? To answer this question, we have to consider the strongest realistic attack vector. In this work we assume that the adversary has access to everything but the missing parameters held by the other party. How easy would it be for this adversary to recover the missing part of the model? We introduce this as the *problem of model completion*:

*Given the entire training data set or an environment simulator, and a subset of the parameters of a trained model, how much training is required to recover the model's original performance?*

In this paper, we define the problem of model completion formally (Section 3.1), propose a metric to measure the hardness of model completion (Section 3.2), and provide empirical results (Section 4 and Section 5) in both the supervised learning (SL) and in reinforcement learning (RL). For our SL experiments we use the AlexNet convolutional network (Krizhevsky et al., 2012) and the ResNet50 residual network (He et al., 2015a) on ImageNet (Deng et al., 2009); for RL we use A3C (Mnih et al., 2015) and Rainbow (Hessel et al., 2017) in the Atari domain (Bellemare et al., 2013) and IMPALA (Espeholt et al., 2018) on DeepMind Lab (Beattie et al., 2016). After training the model, we reinitialize one of the model's layers and measure how much training is required to complete it (see Figure 1).

Our key findings are: (1) Residual networks are easier to complete than nonresidual networks (Figure 3 and Figure 2). (2) For A3C lower layers are often harder to complete than upper layers (Figure 4). (3) The absolute number of parameters has a minimal effect on the hardness of model completion. (4) RL models are harder to complete than SL models. (5) When completing RL models access to the right experience matters (Figure 5).

## 2 RELATED WORK

### 2.1 MODEL COMPLETION

The closest well-studied phenomenon to model completion is *unsupervised pretraining*, first introduced by Hinton et al. (2006). In unsupervised pretraining a subset of the model, typically the lower layers, is trained in a first pass with an unsupervised reconstruction loss (Erhan et al., 2010). The aim is to learn useful high-level representations that make a second pass with a supervised loss more computationally and sample efficient. This second pass could be thought as model completion.

In this paper we study *vertical* model completion where all parameters in one layer have to be completed. Instead we could have studied *horizontal* model completion where some parameters have to be completed in every layer. Horizontal model completion should be easy as suggested by the effectiveness of dropout as a regularizer (Srivastava et al., 2014), which trains a model to be resilient to horizontal parameter loss.

Pruning neural networks (LeCun et al., 1990) is in a sense the reverse operation to model completion. Changpinyo et al. (2017) prune individual connections and Molchanov et al. (2017) prune entire feature maps using different techniques; their findings, lower layers are more important, are compatible with ours. Frankle and Carbin (2018) present empirical evidence for the *lottery ticket*

*hypothesis*: only a small subnetwork matters (the 'lottery ticket') and the rest can be pruned away without loss of performance. The model completion problem for this lottery ticket (which is spread over all layers) would be trivial by definition. All of these works only consider removing parts of the model horizontally.

The model completion problem can also be viewed as transfer learning from one task to the same task, while only sharing a subset of the parameters (Parisotto et al., 2015; Teh et al., 2017). Yosinski et al. (2014) investigate which layers in a deep convolutional model contain general versus task-specific representations; some of their experiments follow the same setup as we do here and their results are in line with ours, but they do not measure the *hardness* of model completion task.

Finally, our work has some connections to distillation of deep models (Bucilua et al., 2006; Hinton et al., 2015; Rusu et al., 2015; Berseth et al., 2018). Distillation can be understood as a 'reverse' of model completion, where we want to find a smaller model with the same performance instead of completing a smaller, partial model.

## 2.2 SHARED MODEL GOVERNANCE

The literature revolves around two techniques for sharing model governance: *homomorphic encryption* (HE; Rivest et al., 1978) and *secure multi-party computation* (MPC; Yao, 1982; Damgård et al., 2012). Both HE and MPC have been successfully applied to machine learning on small datasets like MNIST (Gilad-Bachrach et al., 2016; Mohassel and Zhan, 2017; Dahl, 2017; Wagh et al., 2018) and the Wisconsin Breast Cancer Data set (Graepel et al., 2012).

HE is an encryption scheme that allows computation on encrypted numbers without decrypting them. It thus enables a model to be trained by an untrusted third party in encrypted form. The encryption key to these parameters can be cryptographically shared between several other parties who effectively retain control over how the model is used.

Using MPC numbers can be shared across several parties such that each share individually contains no information about these numbers. Nevertheless computational operations can be performed on the shared numbers if every party performs operations on their share. The result of the computation can be reconstructed by pooling the shares of the result.

While both HE and MPC fulfill a similar purpose, they face different tradeoffs for the additional security benefits: HE incurs a large computational overhead (Lepoint and Naehrig, 2014) while MPC incurs a much smaller computational overhead in exchange for a greater communication overhead (Keller et al., 2016). Moreover, HE provides cryptographic security (reducing attacks to break the cipher on well-studied hard problems such as the discrete logarithm) while MPC provides perfect information-theoretic guarantees as long as the parties involved (3 or more) do not collude.

There are many applications where we would be happy to pay for the additional overhead because we cannot train the model any other way, for example in the health sector where privacy and security are critical. However, if we want to scale shared model governance to the training of large neural networks, both HE and MPC are ruled out because of their prohibitive overhead. In contrast to HE and MPC, sharing governance via model splitting incurs minimal computational and manageable communication overhead. However, instead of strong security guarantees provided by HE and MPC, the security guarantee is bounded from above by the hardness of the model completion problem we study in this paper.

## 3 THE PROBLEM OF MODEL COMPLETION

Let $f_\theta$ be a model parameterized by the vector $\theta$. We consider two settings: supervised learning and reinforcement learning. In our supervised learning experiments we evaluate the model $f_\theta$ by its performance on the test loss $L(\theta)$.

In reinforcement learning an agent interacts with an environment over a number of discrete time steps (Sutton and Barto, 1998): In time step $t$, the agent takes an *action* $a_t$ and receives an *observation* $o_{t+1}$ and a *reward* $r_{t+1} \in \mathbb{R}$ from the environment. We consider the episodic setting in which there is a random final time step $\tau \leq K$ for some constant $K \in \mathbb{N}$, after which we restart with timestep $t = 1$. The agent's goal is to maximize the episodic return $G := \sum_{t=1}^{\tau} r_t$. Its *policy* is a mapping

from sequences of observations to a distribution over actions parameterized by the model $f_\theta$. To unify notation for SL and RL, we equate $L(\theta) = \mathbb{E}_{a_t \sim f_\theta(o_1,\dots,o_{t-1})}[-G]$ such that the loss function for RL is the negative expected episodic return.

## 3.1 PROBLEM DEFINITION

To quantify training costs we measure the computational cost during (re)training. To simplify, we assume that training proceeds over a number of discrete steps. A step can be computation of gradients and parameter update for one minibatch in the case of supervised learning or one environment step in the case of reinforcement learning. We assume that computational cost are constant for each step, which is approximately true in our experiments. This allows us to measure training cost through the number of training steps executed.

Let $T$ denote the *training procedure* for the model $f_\theta$ and let $\theta_0, \theta_1, \dots$ be the sequence of parameter vectors during training where $\theta_i$ denotes the parameters in training step $i$. Furthermore, let $\ell^* := \min\{L(\theta_i) \mid i \leq N\}$ denote the best model performance during the training procedure $T$ (not necessarily the performance of the final weights). We define the *training cost* as the random variable $C_T(\ell) := \arg\min_{i \in \mathbb{N}}\{L(\theta_i) \leq \ell\}$, the number of training steps until the loss falls below the given threshold $\ell \in \mathbb{R}$. After we have trained the model $f_\theta$ for $N$ steps and thus end up with a set of trained parameters $\theta_N$ with loss $L(\theta_N)$, we split the parameters $\theta_N = [\theta_N^1, \theta_N^2]$ into two disjoint subvectors of parameters $\theta_N^1$ and $\theta_N^2$. For example, $\theta_N^2$ could be all parameters of one of the model's layers. The model completion problem is, given the parameters $\theta_N^1$ but not $\theta_N^2$, recovering a model that has loss at most $L(\theta_N)$. This is illustrated in Figure 1.

## 3.2 MEASURING THE HARDNESS OF MODEL COMPLETION

How hard is the model completion problem? To answer this question, we use the parameters $\theta_0' := [\theta_0'^1, \theta_0'^2]$ where $\theta_0'^1 := \theta_N^1$ are the previously trained parameters and $\theta_0'^2$ are freshly initialized parameters. We then execute a (second) *retraining procedure* $T' \in \mathcal{T}$ from a fixed set of available retraining procedures $\mathcal{T}$.[1] The aim of this retraining procedure is to complete the model, and it may be different from the initial training procedure $T$. We assume that $T \in \mathcal{T}$ since retraining the entire model from scratch (reinitializing all parameters) is a valid way to complete the model.

Let $\theta_0', \theta_1', \dots$ be the sequence of parameter vectors obtained from running the retraining procedure $T' \in \mathcal{T}$. Analogously to before, we define $C_{T'}'(\ell) := \arg\min_{i \in \mathbb{N}}\{L(\theta_i') \leq \ell\}$ as the *retraining cost* to get a model whose test loss is below the given threshold $\ell \in \mathbb{R}$. Note that by definition, for $T' = T$ we have that $C_{T'}'(\ell)$ is equal to $C_T(\ell)$ in expectation.

In addition to recovering a model with the best original performance $\ell^*$, we also consider *partial model completion* by using some higher thresholds $\ell_\alpha^* := \alpha\ell^* + (1-\alpha)L(\theta_0)$ for $\alpha \in [0,1]$. These higher thresholds $\ell_\alpha^*$ correspond to the relative progress $\alpha$ from the test loss of the untrained model parameters $L(\theta_0)$ to the best test loss $\ell^*$. Note that $\ell_1^* = \ell^*$.

We define the *hardness of model completion* as the expected cost to complete the model as a fraction of the original training cost for the fastest retraining procedure $T' \in \mathcal{T}$ available:

$$\text{MC-hardness}_T(\alpha) := \inf_{T' \in \mathcal{T}} \mathbb{E}\left[\frac{C_{T'}'(\ell_\alpha^*)}{C_T(\ell_\alpha^*)}\right], \tag{1}$$

where the expectation is taken over all random events in the training procedures $T$ and $T'$.

It is important to emphasize that the hardness of model completion is a *relative* measure, depending on the original training cost $C_T(\ell_\alpha^*)$. This ensures that we can compare the hardness of model completion across different tasks and different domains. In particular, for different values of $\alpha$ we compare like with like: *MC-hardness$_T(\alpha)$ is measured relative to how long it took to get the loss below the threshold $\ell_\alpha^*$ during training*. Importantly, it is *not* relative to how long it took to train the model to its best performance $\ell^*$. This means that naively counter-intuitive results such as MC-hardness$_T(0.8)$ being less than MC-hardness$_T(0.5)$ are possible.

---

[1] $\mathcal{T}$ should not include unrealistic retraining procedures like setting the weights to $\theta_N$ in one step.

Since $C_T(\ell)$ and $C_{T'}(\ell)$ are nonnegative, MC-hardness$_T(\alpha)$ is nonnegative. Moreover, since $T \in \mathcal{T}$ by assumption, we could retrain all model parameters from scratch (formally setting $T'$ to $T$). Thus we have MC-hardness$_T(\alpha) \leq 1$, and therefore MC-hardness is bounded between 0 and 1.

## 3.3 Retraining procedures

Equation 1 denotes an infimum over available retraining procedures $\mathcal{T}$. However, in practice there is a vast number of possible retraining procedures we could use and we cannot enumerate and run all of them. Instead, we take an empirical approach for estimating the hardness of model completion: we investigate the following set of retraining strategies $\mathcal{T}$ to complete the model. All the retraining strategies, if not noted otherwise, are built on top of the original training procedure $T$. Our best result are only an *upper bound* on the hardness of model completion. It is likely that much faster retraining procedures exist.

**T$_1$** *Optimizing $\theta_0'^1$ and $\theta_0'^2$ jointly.* We repeat the original training procedure $T$ on the preserved parameters $\theta_0'^1$ and reinitialized parameters $\theta_0'^2$. The objective function is optimized with respect to all the trainable variables in the model. We might vary in hyperparameters such as learning rates or loss weighting schemes compared to $T$, but keep hyperparameters that change the structure of the model (e.g. size and number of layers) fixed.

**T$_2$** *Optimizing $\theta_0'^2$, but not $\theta_0'^1$.* Similarly to **T$_1$**, in this retraining procedure we keep the previous model structure. However, we freeze the trained weights $\theta_0'^1$, and only train the reinitialized parameters $\theta_0'^2$.

**T$_3$** *Overparametrizing the missing layers.* This builds on retraining procedure **T$_1$**. Overparametrization is a common trick in computer vision, where a model is given a lot more parameters than required, allowing for faster learning. This idea is supported by the 'lottery ticket hypothesis' (Frankle and Carbin, 2018): a larger number of parameters increases the odds of a subpart of the network having random initialization that is more conducive to optimization.

**T$_4$** *Reinitializing parameters $\theta_0'^2$ using a different initialization scheme.* Previous research shows that parameter initialization schemes can have a big impact on convergence properties of deep neural networks (Glorot and Bengio, 2010; Sutskever et al., 2013). In **T$_1$** our parameters are initialized using a *glorot uniform* scheme. This retraining procedure is identical to **T$_1$** except that we reinitialize $\theta_0'^2$ using one of the following weight initialization schemes: *glorot normal* (Glorot and Bengio, 2010), *msra* (He et al., 2015b) or *caffe* (Jia et al., 2014).

## 4 Experimental setup

Our main experimental results establish upper bounds on the hardness of model completion in the context of several state of the art models for both supervised learning and reinforcement learning. In all the experiments, we train a model to a desired performance level (this does not have to be state-of-the-art performance), and then reinitialize a specific part of the network and start the retraining procedure. Each experiment is run with 3 seeds, except IMPALA (5 seeds) and A3C (10 seeds).

**Supervised learning.** We train AlexNet (Krizhevsky et al., 2012) and ResNet50 (He et al., 2015a) on the ImageNet dataset (Deng et al., 2009) to minimize cross-entropy loss. The test loss is the top-1 error rate on the test set. AlexNet is an eight layer convolutional network consisting of five convolutional layers with max-pooling, followed by two fully connected layers and a softmax output layer. ResNet50 is a 50 layer convolutional residual network: The first convolutional layer with max-pooling is followed by four sections, each with a number of ResNet blocks (consisting of two convolutional layers with skip connections and batch normalization), followed by average pooling, a fully connected layer and a softmax output layer. We apply retraining procedures **T$_1$** and **T$_2$** and use a different learning rate schedule than in the original training procedure because it performs better during retraining. All other hyperparameters are kept the same.

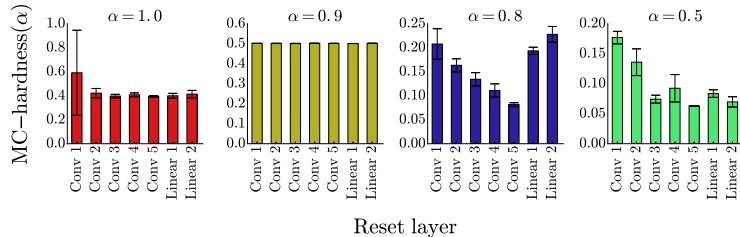

Figure 2: Hardness of model completion for AlexNet on ImageNet under retraining procedure $\mathbf{T_1}$. The x-axis shows experiments that retrain different parts of the model.

**Reinforcement learning.** We consider three different state of the art agents: A3C (Mnih et al., 2016), Rainbow (Hessel et al., 2017) and the IMPALA reinforcement learning agent (Espeholt et al., 2018). A3C comes from a family of actor-critic methods which combine value learning and policy gradient approaches in order to reduce the variance of the gradients. Rainbow is an extension of the standard DQN (Mnih et al., 2015) agent, which combines double Q-learning (van Hasselt, 2010), dueling networks (Wang et al., 2016), distributional RL (Bellemare et al., 2017) and noisy nets (Fortunato et al., 2017). Moreover, it is equipped with a replay buffer that stores the previous million transitions of the form $(o_t, a_t, r_{t+1}, o_{t+1})$, which is then sampled using a prioritized weighting scheme based on temporal difference errors (Schaul et al., 2015). Finally, IMPALA is an extension of A3C, which uses the standard actor-critic architecture with off-policy corrections in order to scale effectively to a large scale distributed setup. We train IMPALA with population based training (Jaderberg et al., 2017).

For A3C and Rainbow we use the Atari 2600 domain (Bellemare et al., 2013) and for IMPALA DeepMind Lab (Beattie et al., 2016). In both cases, we treat the list of games/levels as a single learning problem by averaging across games in Atari and training the agent on all level in parallel in case of DeepMind Lab. In order to reduce the noise in the MC-hardness metric, caused by agents being unable to learn the task and behaving randomly, we filter out the levels in which the original trained agent performs poorly. We apply the retraining procedures $\mathbf{T_1}$, $\mathbf{T_2}$ on all the models, and on A3C we apply additionally $\mathbf{T_3}$ and $\mathbf{T_4}$. All the hyperparameters are kept the same during the training and retraining procedures.

Further details of the training and retraining procedures for all models can be found in Appendix A, and the parameter counts of the layers are listed in Appendix B.

## 5 KEY FINDINGS

Our experimental results on the hardness of the model completion problem are reported in Figures 2–6. These figures show on the x-axis different experiments with different layers being reinitialized (lower to higher layers from left to right). We plot MC-hardness$_T(\alpha)$ as a bar plot with error bars showing the standard deviation over multiple experiment runs with different seeds; the colors indicate different values of $\alpha$. The numbers are provided in Appendix C. In the following we discuss the results.

**1. In the majority of cases, $\mathbf{T_1}$ is the best of our retraining procedures.** From the retraining procedures listed in Section 3.3 we use $\mathbf{T_1}$ and $\mathbf{T_2}$ in all experiments and find that $\mathbf{T_1}$ performs substantially better in all settings except two: First, for A3C, starting from the third convolutional layer, $\mathbf{T_2}$ has lower MC-hardness for all the threshold levels (Figure 4). Second, $\mathbf{T_2}$ performs well on all the layers when retraining ResNet-50, for all $\alpha \leq 0.9$ (Figure 3); the difference is especially visible at $\alpha = 0.9$.

For A3C we use all four retraining procedures. The difference between $\mathbf{T_1}$ and $\mathbf{T_2}$ are shown in Figure 4. For $\mathbf{T_3}$ we tried replacing the first convolutional layer with two convolutional layers using a different kernel size, as well as replacing a fully connected layer with two fully connected layers of varying sizes. The results were worse than using the same architecture and we were often unable to retrieve $100\%$ of the original performance. With $\mathbf{T_4}$ we do not see any statistically significant difference in retraining time between the initialization schemes *glorot normal*, *msra*, and *caffe*.

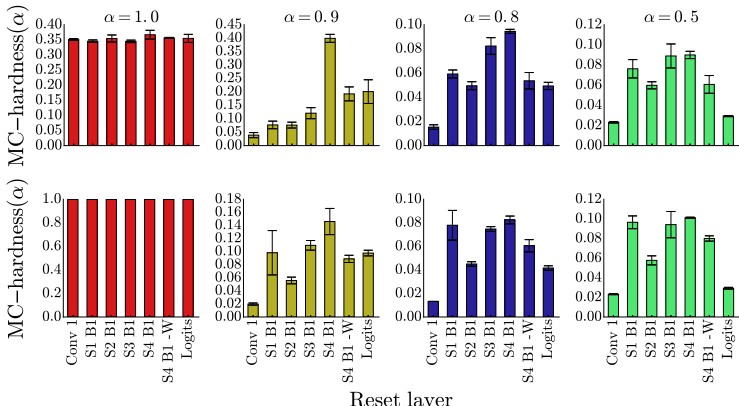

Figure 3: Hardness of model completion for ResNet50 on ImageNet under retraining procedure $\mathbf{T_1}$ (top) and $\mathbf{T_2}$ (bottom). The x-axis shows experiments that retrain different parts of the model where S corresponds to a ResNet section and B corresponds to a block in that section. S4 B1 -W is the same as S4 B1 except that the skip connection does not get reinitialized.

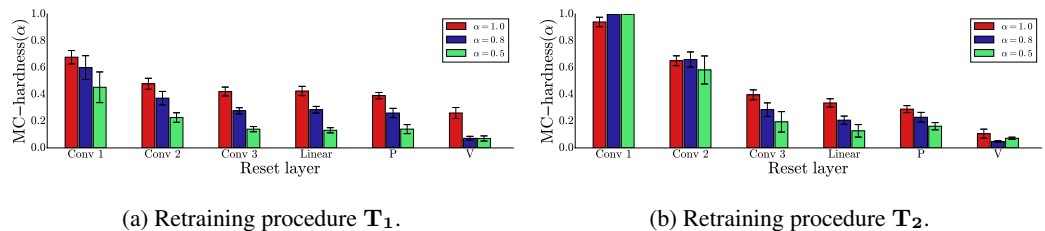

(a) Retraining procedure $\mathbf{T_1}$.        (b) Retraining procedure $\mathbf{T_2}$.

Figure 4: A3C on Atari, trained for 50m steps. For each of 10 sseeds MC-hardness is averaged over 44 Atari games.

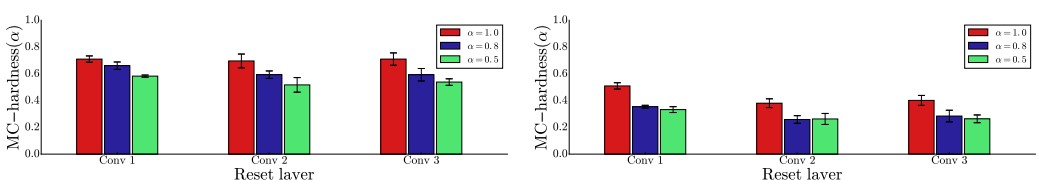

Figure 5: Rainbow on Atari, trained for 5m steps and using retraining procedure $\mathbf{T_1}$. The replay buffer is either reset before retraining (left) or kept intact (right). For each of 3 seeds MC-hardness is averaged over 54 Atari games.

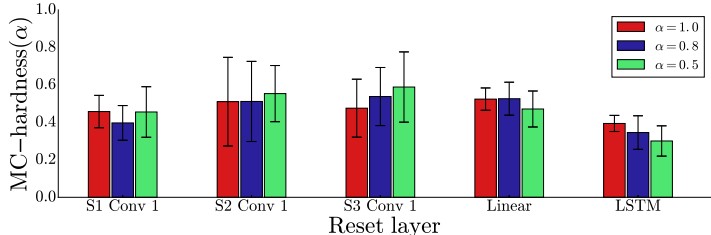

Figure 6: IMPALA on DeepMind Lab with retraining procedure $\mathbf{T_1}$. Each of 5 seeds shows the MC-hardness of a single agent jointly trained on 28 DeepMind Lab levels for a total of 1 billion steps.

**2. Residual networks are easier to complete.** Comparing our SL results in Figure 2 and Figure 3 for $\mathbf{T_1}$, the model hardness for threshold $\alpha = 0.5$ and $\alpha = 0.8$ is much lower for ResNet50 than for AlexNet. However, to get the original model performance ($\alpha = 1$), both models need about 40% of the original training cost. As mentioned above, $\mathbf{T_2}$ works better than $\mathbf{T_1}$ on ResNet50 for $\alpha \le 0.9$.

An intact skip connection helps retraining for $\alpha \le 0.9$ and $\mathbf{T_1}$, but not $\mathbf{T_2}$, as illustrated in the experiment S4 B1 -W (Figure 3). A noticeable outlier is S4 B1 at $\alpha = 0.9$; it is unclear what causes this effect, but it reproduced every time we ran this experiment.

Residual neural networks use skip connections across two or more layers (He et al., 2015a). This causes the features in those layers to be additive with respect to the incoming features, rather than replacing them as in non-residual networks. Thus lower-level and higher-level representations tend to be more spread out across the network, rather than being confined to lower and higher layers, respectively. This would explain why model completion in residual networks is more independent of the location of the layer.

**3. For A3C lower layers are often harder to complete than upper layers.** Figure 4 shows that for A3C the lower layers are harder to complete than the higher layers since for each value of $\alpha$ the MC-hardness decreases from left to right. However, this effect is much smaller for Rainbow (Figure 5) and AlexNet (Figure 2).

In nonresidual networks lower convolutional layers typically learn much simpler and more general features that are more task independent (Yosinski et al., 2014). Moreover, noise perturbations of lower layers have a significantly higher impact on the performance of deep learning models since noise grows exponentially through the network layers (Raghu et al., 2016). Higher level activations are functions of the lower level ones; if a lower layer is reset, all subsequent activations will be invalidated. This could imply that the gradients on the higher layers are incorrect and thus slow down training.

**4. The absolute number of parameters has a minimal effect on the hardness of model completion.** If information content is spread uniformly across the model, then model completion should be a linear function in the number of parameters that we remove. However, the number of parameters in deep models usually vary greatly between layers; the lower-level convolutional layers have 2–3 orders of magnitude fewer parameters than the higher level fully connected layers and LSTMs (see Appendix B).

In order to test this explicitly, we performed an experiment on AlexNet both increasing and decreasing the total number of feature maps and fully connected units in every layer by 50%, resulting in approximately an order of magnitude difference in terms of parameters between the two models. We found that there is no significant difference in MC-hardness across all threshold levels.

**5. RL models are harder to complete than SL models.** Across all of our experiments, the model completion of individual layers for threshold $\alpha = 1$ in SL (Figure 2 and Figure 3) is easier than the model completion in RL (Figure 4, Figure 5, and Figure 6). In many cases the same holds from lower thresholds as well.

By resetting one layer of the model we lose access to the agent's ability to generate useful experience from interaction with the environment. As we retrain the model, the agent has to re-explore the environment to gather the right experience again, which takes extra training time. While this effect is also present during the training procedure $T$, it is possible that resetting one layer makes the exploration problem harder than acting from a randomly initialized network.

**6. When completing RL models access to the right experience matters.**   To understand this effect better, we allow the retraining procedure access to Rainbow's replay buffer. At the start of retraining this replay buffer is filled with experience from the fully trained policy. Figure 5 shows that the model completion hardness becomes much easier with access to this replay buffer: the three left bar plots are lower than the three right.

This result is supported by the benefits of kickstarting (Schmitt et al., 2018), where a newly trained agent gets access to an expert agent's policy. Moreover, this is consistent with findings by Hester et al. (2018), who show performance benefits by adding expert trajectories to the replay buffer.

## 6 DISCUSSION

Our results shed some initial glimpse on the model completion problem and its hardness. Our findings include: residual networks are easier to complete than non-residual networks, lower layers are often harder to complete than higher layers, and RL models are harder to complete than SL models. Nevertheless several question remain unanswered: Why is the difference in MC-hardness less pronounced between lower and higher layers in Rainbow and AlexNet than in A3C? Why is the absolute number of parameters insubstantial? Are there retraining procedures that are faster than $\mathbf{T_1}$?

Furthermore, our definition of hardness of the model completion problem creates an opportunity to *modulate* the hardness of model completion. For example, we could devise model architectures with the explicit objective that model completion be easy (to encourage robustness) or hard (to increase security when sharing governance through model splitting). Importantly, since Equation 1 can be evaluated automatically, we can readily combine this with architecture search (Zoph and Le, 2017).

Our experiments show that when we want to recover $100\%$ of the original performance, model completion may be quite costly: $\sim 40\%$ of the original training costs in many settings; lower performance levels often retrain significantly faster. In scenarios where a model gets trained over many months or years, $40\%$ of the cost may be prohibitively expensive. However, this number also has to be taken with a grain of salt because there are many possible retraining procedures that we did not try. The security properties of model splitting as a method for shared governance require further investigation: in addition to more effective retraining procedures, an attacker may also have access to previous activations or be able to inject their own training data. Yet our experiments suggest that model splitting could be a promising method for shared governance. In contrast to MPC and HE it has a substantial advantage because it is cost-competitiveness with normal training and inference.

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

| Learning rate | Training batches | Retraining batches |
|---|---|---|
| $5e-2$ | 0 | 0 |
| $5e-3$ | $60e3$ | $30e3$ |
| $5e-4$ | $90e3$ | $45e3$ |
| $5e-5$ | $105e3$ | $72.5e3$ |

Table 1: AlexNet: Learning schedule for training and retraining procedures.

| Learning rate | Training batches | Retraining batches |
|---|---|---|
| $1e-1$ | 0 | / |
| $1e-2$ | $30e3$ | 0 |
| $1e-3$ | $45e3$ | $20e3$ |

Table 2: ResNet50: Learning schedule for training and retraining procedures.

## A  EXPERIMENTAL DETAILS

**AlexNet**   We train this model for 120e3 batches, with batch size of 256. We apply batch normalization on the convolutional layers and $\ell_2$-regularization of 1e-4. Optimization is done using Momentum SGD with momentum of 0.9 and the learning rate schedule which is shown in Table 1. Note that the learning schedule during retraining is 50% faster than during training (for $\mathbf{T_1}$ and $\mathbf{T_2}$).

For both retraining procedures $\mathbf{T_1}$ and $\mathbf{T_2}$, we perform reset for each of the first 5 convolutional layers and the following 2 fully connected layers. Table 3 shows the number of trainable parameters for each of the layers.

**ResNet50**   We perform all training and retraining procedures for 60e3 batches, with batch size of 64 and $\ell_2$-regularization of 1e-4. Optimization is done using Momentum SGD with momentum of 0.9 and the learning rate schedule shown in Table 2.

For our experiments, we reinitialize the very first convolutional layer, as well as the first ResNet block for each of the four subsequent network sections. In the 'S4 B1 -W' experiment, we leave out resetting the learned skip connection. Finally, we also reset the last fully connected layer containing logits.

**A3C**   Each agent is trained on a single Atari level for 5e7 environment steps, over 10 seeds. We use the standard Atari architecture consisting of 3 convolutional layers, 1 fully connected layer and 2 fully connected 'heads' for policy and value function. The number of parameters for each of those layers is shown in Table 5. For optimization, we use RMSProp optimizer with $\epsilon = 0.1$, decay of 0.99 and $\alpha = 6e$-4 that is linearly annealed to 0. For all the other hyperparameters we refer to Mnih et al. (2016). Finally, while calculating reported statistics we removed the following Atari levels, due to poor behaviour of the trained agent: Montezuma's Revenge, Venture, Solaris, Enduro, Battle Zone, Gravitar, Kangaroo, Skiing, Krull, Video pinball, Freeway, Centipede, and Robotank.

**Rainbow**   Each agent is trained on a single Atari level for 20e6 environment frames, over 3 seeds. Due to agent behaving randomly, we remove the following Atari games from our MC-hardness calculations: Montezuma's Revenge, Venture, and Solaris. For our experiments, we use the same network architecture and hyperparameters as reported in Hessel et al. (2017) and target the first 3 convolutional layers. Table 6 has the total number of parameters for each of the 3 layers.

**IMPALA**   We train a single agent over a suite of 28 DeepMind Lab levels for a total of 1 billion steps over all the environments, over 5 seeds. During training we apply population based training (PBT; Jaderberg et al., 2017) with population of size 12, in order to evolve the entropy cost, learning rate and $\epsilon$ for RMSProp.  For language modelling a separated LSTM channel is used.  In the results we report, we removed two DeepMind Lab levels due to poor behavior of the trained agent: 'language_execute_random_task' and 'psychlab_visual_search'. All the other hyperparameters are retained from Espeholt et al. (2018). For our experiments, we reinitialize the first convolutional layer

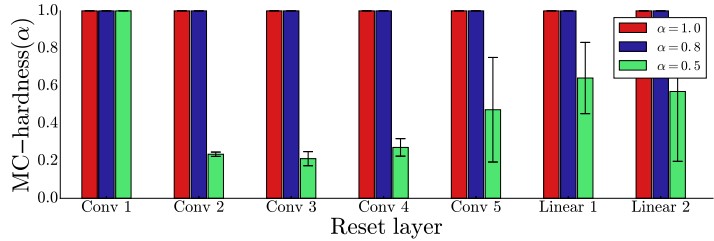

Figure 7: Hardness of model completion for AlexNet on ImageNet under retraining procedure $\mathbf{T_2}$. The x-axis shows experiments that retrain different parts of the model.

in each of the 3 ResNet sections, as well as the final fully connected layer and the weights of the LSTM. The absolute number of parameters for each of the aforementioned layers is available in Table 7.

## B    PARAMETER COUNTS

|  | Conv 1 | Conv 2 | Conv 3 | Conv 4 | Conv 5 | Linear 1 | Linear 2 | Logits | Total |
|---|---|---|---|---|---|---|---|---|---|
| **Standard** | 34,944 | 614,654 | 885,120 | 1,327,488 | 884,992 | 1,052,672 | 16,781,312 | 4,096,000 | 25,677,182 |
| **Large** | 52,416 | 1,382,784 | 1,991,232 | 2,986,560 | 1,991,000 | 2,365,440 | 37,754,880 | 6,144,000 | 54,668,312 |
| **Small** | 17,472 | 153,728 | 221,376 | 331,968 | 221,312 | 264,192 | 4,196,352 | 2,048,000 | 7,454,400 |

Table 3: AlexNet, number of parameters per layer. Compared to the 'Standard' version, 'Small' represents a network with 50% less feature maps and fully connected neurons, whereas 'Large' represents a model with 50% more feature maps and fully connected neurons.

|  | Conv1 | S1 B1 | S2 B1 | S3 B1 | S4 B1 | S4 B1 -W | Logits |
|---|---|---|---|---|---|---|---|
| **ResNet50** | 9,536 | 75,008 | 379,392 | 1,512,448 | 6,039,552 | 3,938,304 | 2,048,000 |

Table 4: ResNet50, number of parameters per layer.

|  | Conv 1 | Conv 2 | Conv 3 | Linear | Policy head | Value head | Total |
|---|---|---|---|---|---|---|---|
| **A3C** | 8,192 | 32,768 | 36,864 | 1,179,648 | 9,216 | 512 | 1,267,200 |

Table 5: A3C, number of parameters per layer.

|  | Conv 1 | Conv 2 | Conv 3 | Total |
|---|---|---|---|---|
| **Rainbow** | 8,192 | 32,768 | 36,864 | 1,267,712 |

Table 6: Rainbow, number of parameters per layer.

|  | S1 Conv 1 | S2 Conv 1 | S3 Conv 1 | Linear | LSTM |
|---|---|---|---|---|---|
| **IMPALA** | 432 | 4,608 | 9,216 | 3,538,944 | 526,336 |

Table 7: IMPALA reinforcement learning agent, number of parameters per layer.

# C EXPERIMENTAL DATA

## C.1 ALEXNET

**MC-hardness($\alpha = 1.0$)**

| Layer | Retraining procedure | Median | Mean | SD | Min | Max |
|-------|---------------------|--------|------|------|------|------|
| Conv 1 | $T_1$ | 0.39 | 0.59 | 0.35 | 0.38 | 1.00 |
|        | $T_2$ | 1.00 | 1.00 | 0.00 | 1.00 | 1.00 |
| Conv 2 | $T_1$ | 0.44 | 0.42 | 0.04 | 0.38 | 0.45 |
|        | $T_2$ | 1.00 | 1.00 | 0.00 | 1.00 | 1.00 |
| Conv 3 | $T_1$ | 0.40 | 0.40 | 0.01 | 0.38 | 0.41 |
|        | $T_2$ | 1.00 | 1.00 | 0.00 | 1.00 | 1.00 |
| Conv 4 | $T_1$ | 0.41 | 0.41 | 0.02 | 0.39 | 0.42 |
|        | $T_2$ | 1.00 | 1.00 | 0.00 | 1.00 | 1.00 |
| Conv 5 | $T_1$ | 0.39 | 0.39 | 0.01 | 0.39 | 0.40 |
|        | $T_2$ | 1.00 | 1.00 | 0.00 | 1.00 | 1.00 |
| Linear 1 | $T_1$ | 0.39 | 0.40 | 0.02 | 0.39 | 0.42 |
|          | $T_2$ | 1.00 | 1.00 | 0.00 | 1.00 | 1.00 |
| Linear 2 | $T_1$ | 0.41 | 0.41 | 0.03 | 0.38 | 0.44 |
|          | $T_2$ | 1.00 | 1.00 | 0.00 | 1.00 | 1.00 |

**MC-hardness($\alpha = 0.9$)**

| Layer | Retraining procedure | Median | Mean | SD | Min | Max |
|-------|---------------------|--------|------|------|------|------|
| Conv 1 | $T_1$ | 0.5 | 0.5 | 0.0 | 0.5 | 0.5 |
|        | $T_2$ | 1.0 | 1.0 | 0.0 | 1.0 | 1.0 |
| Conv 2 | $T_1$ | 0.5 | 0.5 | 0.0 | 0.5 | 0.5 |
|        | $T_2$ | 1.0 | 1.0 | 0.0 | 1.0 | 1.0 |
| Conv 3 | $T_1$ | 0.5 | 0.5 | 0.0 | 0.5 | 0.5 |
|        | $T_2$ | 1.0 | 1.0 | 0.0 | 1.0 | 1.0 |
| Conv 4 | $T_1$ | 0.5 | 0.5 | 0.0 | 0.5 | 0.5 |
|        | $T_2$ | 1.0 | 1.0 | 0.0 | 1.0 | 1.0 |
| Conv 5 | $T_1$ | 0.5 | 0.5 | 0.0 | 0.5 | 0.5 |
|        | $T_2$ | 1.0 | 1.0 | 0.0 | 1.0 | 1.0 |
| Linear 1 | $T_1$ | 0.5 | 0.5 | 0.0 | 0.5 | 0.5 |
|          | $T_2$ | 1.0 | 1.0 | 0.0 | 1.0 | 1.0 |
| Linear 2 | $T_1$ | 0.5 | 0.5 | 0.0 | 0.5 | 0.5 |
|          | $T_2$ | 1.0 | 1.0 | 0.0 | 1.0 | 1.0 |

**MC-hardness($\alpha = 0.8$)**

| Layer | Retraining procedure | Median | Mean | SD | Min | Max |
|---|---|---|---|---|---|---|
| Conv 1 | $T_1$ | 0.20 | 0.21 | 0.03 | 0.18 | 0.24 |
| | $T_2$ | 1.00 | 1.00 | 0.00 | 1.00 | 1.00 |
| Conv 2 | $T_1$ | 0.17 | 0.16 | 0.01 | 0.15 | 0.17 |
| | $T_2$ | 1.00 | 1.00 | 0.00 | 1.00 | 1.00 |
| Conv 3 | $T_1$ | 0.13 | 0.13 | 0.01 | 0.12 | 0.15 |
| | $T_2$ | 1.00 | 1.00 | 0.00 | 1.00 | 1.00 |
| Conv 4 | $T_1$ | 0.11 | 0.11 | 0.01 | 0.10 | 0.13 |
| | $T_2$ | 1.00 | 1.00 | 0.00 | 1.00 | 1.00 |
| Conv 5 | $T_1$ | 0.08 | 0.08 | 0.00 | 0.08 | 0.08 |
| | $T_2$ | 1.00 | 1.00 | 0.00 | 1.00 | 1.00 |
| Linear 1 | $T_1$ | 0.19 | 0.19 | 0.01 | 0.19 | 0.20 |
| | $T_2$ | 1.00 | 1.00 | 0.00 | 1.00 | 1.00 |
| Linear 2 | $T_1$ | 0.22 | 0.23 | 0.02 | 0.21 | 0.25 |
| | $T_2$ | 1.00 | 1.00 | 0.00 | 1.00 | 1.00 |

**MC-hardness($\alpha = 0.5$)**

| Layer | Retraining procedure | Median | Mean | SD | Min | Max |
|---|---|---|---|---|---|---|
| Conv 1 | $T_1$ | 0.18 | 0.18 | 0.01 | 0.17 | 0.19 |
| | $T_2$ | 1.00 | 1.00 | 0.00 | 1.00 | 1.00 |
| Conv 2 | $T_1$ | 0.13 | 0.14 | 0.02 | 0.12 | 0.16 |
| | $T_2$ | 0.24 | 0.24 | 0.01 | 0.22 | 0.24 |
| Conv 3 | $T_1$ | 0.08 | 0.07 | 0.01 | 0.07 | 0.08 |
| | $T_2$ | 0.20 | 0.21 | 0.04 | 0.18 | 0.25 |
| Conv 4 | $T_1$ | 0.08 | 0.09 | 0.02 | 0.08 | 0.12 |
| | $T_2$ | 0.26 | 0.27 | 0.05 | 0.23 | 0.32 |
| Conv 5 | $T_1$ | 0.06 | 0.06 | 0.00 | 0.06 | 0.06 |
| | $T_2$ | 0.41 | 0.47 | 0.28 | 0.23 | 0.78 |
| Linear 1 | $T_1$ | 0.08 | 0.08 | 0.01 | 0.08 | 0.09 |
| | $T_2$ | 0.70 | 0.64 | 0.19 | 0.43 | 0.79 |
| Linear 2 | $T_1$ | 0.07 | 0.07 | 0.01 | 0.06 | 0.08 |
| | $T_2$ | 0.37 | 0.57 | 0.37 | 0.34 | 1.00 |

## C.2 RESNET50

**MC-hardness($\alpha = 1.0$)**

| Layer | Retraining procedure | Median | Mean | SD | Min | Max |
|---|---|---|---|---|---|---|
| Conv 1 | $T_1$ | 0.35 | 0.35 | 0.00 | 0.35 | 0.35 |
| | $T_2$ | 1.00 | 1.00 | 0.00 | 1.00 | 1.00 |
| S1 B1 | $T_1$ | 0.35 | 0.35 | 0.00 | 0.34 | 0.35 |
| | $T_2$ | 1.00 | 1.00 | 0.00 | 1.00 | 1.00 |
| S2 B1 | $T_1$ | 0.35 | 0.35 | 0.01 | 0.34 | 0.37 |
| | $T_2$ | 1.00 | 1.00 | 0.00 | 1.00 | 1.00 |
| S3 B1 | $T_1$ | 0.34 | 0.34 | 0.00 | 0.34 | 0.35 |
| | $T_2$ | 1.00 | 1.00 | 0.00 | 1.00 | 1.00 |
| S4 B1 | $T_1$ | 0.36 | 0.37 | 0.01 | 0.36 | 0.38 |
| | $T_2$ | 1.00 | 1.00 | 0.00 | 1.00 | 1.00 |
| S4 B1 -W | $T_1$ | 0.36 | 0.36 | 0.00 | 0.35 | 0.36 |
| | $T_2$ | 1.00 | 1.00 | 0.00 | 1.00 | 1.00 |
| Logits | $T_1$ | 0.35 | 0.35 | 0.01 | 0.35 | 0.37 |
| | $T_2$ | 1.00 | 1.00 | 0.00 | 1.00 | 1.00 |

**MC-hardness($\alpha = 0.9$)**

| Layer | Retraining procedure | Median | Mean | SD | Min | Max |
|---|---|---|---|---|---|---|
| Conv 1 | $T_1$ | 0.04 | 0.04 | 0.01 | 0.03 | 0.05 |
| | $T_2$ | 0.02 | 0.02 | 0.00 | 0.02 | 0.02 |
| S1 B1 | $T_1$ | 0.07 | 0.08 | 0.01 | 0.07 | 0.09 |
| | $T_2$ | 0.08 | 0.10 | 0.03 | 0.07 | 0.14 |
| S1 B2 | $T_1$ | 0.07 | 0.08 | 0.01 | 0.07 | 0.09 |
| | $T_2$ | 0.05 | 0.06 | 0.01 | 0.05 | 0.06 |
| S1 B3 | $T_1$ | 0.13 | 0.12 | 0.02 | 0.10 | 0.14 |
| | $T_2$ | 0.11 | 0.11 | 0.01 | 0.10 | 0.12 |
| S1 B4 | $T_1$ | 0.39 | 0.40 | 0.01 | 0.39 | 0.42 |
| | $T_2$ | 0.15 | 0.15 | 0.02 | 0.12 | 0.16 |
| S1 B4 -W | $T_1$ | 0.18 | 0.19 | 0.03 | 0.17 | 0.22 |
| | $T_2$ | 0.09 | 0.09 | 0.01 | 0.08 | 0.09 |
| Logits | $T_1$ | 0.20 | 0.20 | 0.04 | 0.16 | 0.24 |
| | $T_2$ | 0.10 | 0.10 | 0.00 | 0.09 | 0.10 |

**MC-hardness($\alpha = 0.8$)**

| Layer | Retraining procedure | Median | Mean | SD | Min | Max |
|---|---|---|---|---|---|---|
| Conv 1 | $T_1$ | 0.02 | 0.02 | 0.00 | 0.01 | 0.02 |
| | $T_2$ | 0.01 | 0.01 | 0.00 | 0.01 | 0.01 |
| S1 B1 | $T_1$ | 0.06 | 0.06 | 0.00 | 0.06 | 0.06 |
| | $T_2$ | 0.07 | 0.08 | 0.01 | 0.07 | 0.09 |
| S1 B2 | $T_1$ | 0.05 | 0.05 | 0.00 | 0.05 | 0.05 |
| | $T_2$ | 0.05 | 0.04 | 0.00 | 0.04 | 0.05 |
| S1 B3 | $T_1$ | 0.08 | 0.08 | 0.01 | 0.08 | 0.09 |
| | $T_2$ | 0.08 | 0.07 | 0.00 | 0.07 | 0.08 |
| S1 B4 | $T_1$ | 0.10 | 0.09 | 0.00 | 0.09 | 0.10 |
| | $T_2$ | 0.08 | 0.08 | 0.00 | 0.08 | 0.09 |
| S1 B4 -W | $T_1$ | 0.06 | 0.05 | 0.01 | 0.05 | 0.06 |
| | $T_2$ | 0.06 | 0.06 | 0.01 | 0.06 | 0.07 |
| Logits | $T_1$ | 0.05 | 0.05 | 0.00 | 0.05 | 0.05 |
| | $T_2$ | 0.04 | 0.04 | 0.00 | 0.04 | 0.04 |

**MC-hardness($\alpha = 0.5$)**

| Layer | Retraining procedure | Median | Mean | SD | Min | Max |
|---|---|---|---|---|---|---|
| Conv 1 | $T_1$ | 0.02 | 0.02 | 0.00 | 0.02 | 0.02 |
| | $T_2$ | 0.02 | 0.02 | 0.00 | 0.02 | 0.02 |
| S1 B1 | $T_1$ | 0.07 | 0.08 | 0.01 | 0.07 | 0.09 |
| | $T_2$ | 0.10 | 0.10 | 0.01 | 0.09 | 0.10 |
| S1 B2 | $T_1$ | 0.06 | 0.06 | 0.00 | 0.06 | 0.06 |
| | $T_2$ | 0.06 | 0.06 | 0.00 | 0.05 | 0.06 |
| S1 B3 | $T_1$ | 0.09 | 0.09 | 0.01 | 0.08 | 0.10 |
| | $T_2$ | 0.09 | 0.09 | 0.01 | 0.08 | 0.11 |
| S1 B4 | $T_1$ | 0.09 | 0.09 | 0.00 | 0.09 | 0.09 |
| | $T_2$ | 0.10 | 0.10 | 0.00 | 0.10 | 0.10 |
| S1 B4 -W | $T_1$ | 0.06 | 0.06 | 0.01 | 0.05 | 0.07 |
| | $T_2$ | 0.08 | 0.08 | 0.00 | 0.08 | 0.08 |
| Logits | $T_1$ | 0.03 | 0.03 | 0.00 | 0.03 | 0.03 |
| | $T_2$ | 0.03 | 0.03 | 0.00 | 0.03 | 0.03 |

## C.3  A3C

### C.3.1  RETRAINING PROCEDURE $\mathbf{T_1}$

| MC-hardness($\alpha = 1.0$) | | | | | |
|---|---|---|---|---|---|
| | Median | Mean | SD | Min | Max |
| Layer | | | | | |
| Conv 1 | 0.92 | 0.94 | 0.03 | 0.91 | 1.00 |
| Conv 2 | 0.65 | 0.65 | 0.04 | 0.60 | 0.72 |
| Conv 3 | 0.39 | 0.40 | 0.04 | 0.35 | 0.45 |
| Linear 1 | 0.33 | 0.34 | 0.03 | 0.29 | 0.39 |
| P | 0.29 | 0.29 | 0.03 | 0.25 | 0.35 |
| V | 0.11 | 0.11 | 0.03 | 0.06 | 0.16 |

| MC-hardness($\alpha = 0.8$) | | | | | |
|---|---|---|---|---|---|
| | Median | Mean | SD | Min | Max |
| Layer | | | | | |
| Conv 1 | 1.00 | 1.00 | 0.00 | 1.00 | 1.00 |
| Conv 2 | 0.66 | 0.66 | 0.06 | 0.57 | 0.76 |
| Conv 3 | 0.28 | 0.29 | 0.05 | 0.19 | 0.35 |
| Linear 1 | 0.20 | 0.21 | 0.03 | 0.18 | 0.27 |
| P | 0.22 | 0.23 | 0.04 | 0.18 | 0.29 |
| V | 0.05 | 0.05 | 0.01 | 0.04 | 0.06 |

| MC-hardness($\alpha = 0.5$) | | | | | |
|---|---|---|---|---|---|
| | Median | Mean | SD | Min | Max |
| Layer | | | | | |
| Conv 1 | 1.00 | 1.00 | 0.00 | 1.00 | 1.00 |
| Conv 2 | 0.57 | 0.58 | 0.10 | 0.43 | 0.77 |
| Conv 3 | 0.18 | 0.19 | 0.08 | 0.12 | 0.34 |
| Linear 1 | 0.11 | 0.13 | 0.05 | 0.09 | 0.24 |
| P | 0.16 | 0.16 | 0.03 | 0.10 | 0.21 |
| V | 0.07 | 0.07 | 0.01 | 0.06 | 0.09 |

### C.3.2  RETRAINING PROCEDURE $\mathbf{T_2}$

| MC-hardness($\alpha = 1.0$) | | | | | |
|---|---|---|---|---|---|
| | Median | Mean | SD | Min | Max |
| Layer | | | | | |
| Conv 1 | 0.66 | 0.68 | 0.05 | 0.62 | 0.76 |
| Conv 2 | 0.47 | 0.48 | 0.04 | 0.43 | 0.57 |
| Conv 3 | 0.42 | 0.42 | 0.03 | 0.36 | 0.47 |
| Linear 1 | 0.42 | 0.43 | 0.03 | 0.39 | 0.52 |
| P | 0.39 | 0.39 | 0.02 | 0.34 | 0.43 |
| V | 0.25 | 0.26 | 0.04 | 0.21 | 0.35 |

**MC-hardness($\alpha = 0.8$)**

| Layer | Median | Mean | SD | Min | Max |
|---|---|---|---|---|---|
| Conv 1 | 0.57 | 0.60 | 0.09 | 0.51 | 0.78 |
| Conv 2 | 0.36 | 0.37 | 0.05 | 0.31 | 0.46 |
| Conv 3 | 0.27 | 0.28 | 0.02 | 0.25 | 0.32 |
| Linear 1 | 0.29 | 0.29 | 0.02 | 0.23 | 0.32 |
| P | 0.26 | 0.26 | 0.03 | 0.21 | 0.34 |
| V | 0.07 | 0.07 | 0.01 | 0.05 | 0.09 |

**MC-hardness($\alpha = 0.5$)**

| Layer | Median | Mean | SD | Min | Max |
|---|---|---|---|---|---|
| Conv 1 | 0.42 | 0.45 | 0.11 | 0.32 | 0.69 |
| Conv 2 | 0.23 | 0.23 | 0.04 | 0.17 | 0.29 |
| Conv 3 | 0.14 | 0.14 | 0.02 | 0.12 | 0.18 |
| Linear 1 | 0.13 | 0.13 | 0.02 | 0.11 | 0.16 |
| P | 0.14 | 0.14 | 0.03 | 0.09 | 0.19 |
| V | 0.06 | 0.07 | 0.02 | 0.06 | 0.12 |

## C.4 RAINBOW

**MC-hardness($\alpha = 1.0$)**

| Layer | Retraining procedure | Reset Buffer | Median | Mean | SD | Min | Max |
|---|---|---|---|---|---|---|---|
| Conv 1 | $T_1$ | False | 0.52 | 0.51 | 0.02 | 0.48 | 0.53 |
| | | True | 0.71 | 0.71 | 0.02 | 0.69 | 0.73 |
| | $T_2$ | False | 0.90 | 0.89 | 0.03 | 0.86 | 0.91 |
| | | True | 1.00 | 1.00 | 0.00 | 1.00 | 1.00 |
| Conv 2 | $T_1$ | False | 0.39 | 0.38 | 0.03 | 0.34 | 0.41 |
| | | True | 0.70 | 0.70 | 0.05 | 0.64 | 0.75 |
| | $T_2$ | False | 0.63 | 0.66 | 0.06 | 0.63 | 0.74 |
| | | True | 1.00 | 1.00 | 0.00 | 1.00 | 1.00 |
| Conv 3 | $T_1$ | False | 0.39 | 0.40 | 0.04 | 0.37 | 0.44 |
| | | True | 0.70 | 0.71 | 0.05 | 0.67 | 0.76 |
| | $T_2$ | False | 0.59 | 0.59 | 0.03 | 0.55 | 0.61 |
| | | True | 0.88 | 0.87 | 0.03 | 0.83 | 0.89 |

**MC-hardness($\alpha = 0.8$)**

| Layer | Retraining procedure | Reset Buffer | Median | Mean | SD | Min | Max |
|---|---|---|---|---|---|---|---|
| Conv 1 | $T_1$ | False | 0.35 | 0.35 | 0.01 | 0.34 | 0.36 |
| | | True | 0.66 | 0.66 | 0.03 | 0.63 | 0.69 |
| | $T_2$ | False | 0.93 | 0.94 | 0.04 | 0.90 | 0.99 |
| | | True | 1.00 | 1.00 | 0.00 | 1.00 | 1.00 |
| Conv 2 | $T_1$ | False | 0.26 | 0.26 | 0.03 | 0.23 | 0.28 |
| | | True | 0.59 | 0.59 | 0.03 | 0.57 | 0.62 |
| | $T_2$ | False | 0.56 | 0.60 | 0.11 | 0.52 | 0.73 |
| | | True | 1.00 | 1.00 | 0.00 | 1.00 | 1.00 |
| Conv 3 | $T_1$ | False | 0.29 | 0.28 | 0.04 | 0.24 | 0.32 |
| | | True | 0.61 | 0.59 | 0.05 | 0.54 | 0.63 |
| | $T_2$ | False | 0.47 | 0.46 | 0.02 | 0.43 | 0.48 |
| | | True | 0.92 | 0.94 | 0.05 | 0.90 | 1.00 |

| MC-hardness($\alpha = 0.5$) | | | Median | Mean | SD | Min | Max |
|---|---|---|---|---|---|---|---|
| Layer | Retraining procedure | Reset Buffer | | | | | |
| Conv 1 | $T_1$ | False | 0.34 | 0.33 | 0.02 | 0.31 | 0.35 |
| | | True | 0.58 | 0.58 | 0.01 | 0.58 | 0.59 |
| | $T_2$ | False | 1.00 | 1.00 | 0.00 | 1.00 | 1.00 |
| | | True | 1.00 | 1.00 | 0.00 | 1.00 | 1.00 |
| Conv 2 | $T_1$ | False | 0.25 | 0.26 | 0.04 | 0.23 | 0.31 |
| | | True | 0.49 | 0.52 | 0.05 | 0.48 | 0.58 |
| | $T_2$ | False | 0.64 | 0.65 | 0.10 | 0.56 | 0.76 |
| | | True | 1.00 | 1.00 | 0.00 | 1.00 | 1.00 |
| Conv 3 | $T_1$ | False | 0.28 | 0.26 | 0.03 | 0.23 | 0.28 |
| | | True | 0.53 | 0.54 | 0.02 | 0.52 | 0.57 |
| | $T_2$ | False | 0.50 | 0.51 | 0.02 | 0.50 | 0.53 |
| | | True | 1.00 | 1.00 | 0.00 | 1.00 | 1.00 |

## C.5 IMPALA REINFORCEMENT LEARNING AGENT

| MC-hardness($\alpha = 1.0$) | | Median | Mean | SD | Min | Max |
|---|---|---|---|---|---|---|
| Layer | Retraining procedure | | | | | |
| S1 Conv 1 | $T_1$ | 0.44 | 0.46 | 0.09 | 0.38 | 0.59 |
| | $T_2$ | 0.65 | 0.70 | 0.20 | 0.50 | 1.00 |
| S2 Conv 1 | $T_1$ | 0.41 | 0.51 | 0.24 | 0.26 | 0.84 |
| | $T_2$ | 0.65 | 0.63 | 0.26 | 0.37 | 1.00 |
| S3 Conv 1 | $T_1$ | 0.51 | 0.48 | 0.15 | 0.22 | 0.60 |
| | $T_2$ | 0.51 | 0.50 | 0.25 | 0.15 | 0.86 |
| Linear | $T_1$ | 0.49 | 0.52 | 0.06 | 0.48 | 0.60 |
| | $T_2$ | 0.67 | 0.71 | 0.18 | 0.53 | 0.91 |
| LSTM | $T_1$ | 0.39 | 0.39 | 0.04 | 0.34 | 0.46 |
| | $T_2$ | 0.41 | 0.48 | 0.16 | 0.34 | 0.73 |

| MC-hardness($\alpha = 0.9$) | | Median | Mean | SD | Min | Max |
|---|---|---|---|---|---|---|
| Layer | Retraining procedure | | | | | |
| S1 Conv 1 | $T_1$ | 0.41 | 0.40 | 0.09 | 0.25 | 0.47 |
| | $T_2$ | 0.66 | 0.71 | 0.20 | 0.47 | 1.00 |
| S2 Conv 1 | $T_1$ | 0.50 | 0.51 | 0.21 | 0.29 | 0.83 |
| | $T_2$ | 0.70 | 0.66 | 0.26 | 0.31 | 1.00 |
| S3 Conv 1 | $T_1$ | 0.58 | 0.54 | 0.16 | 0.27 | 0.66 |
| | $T_2$ | 0.55 | 0.55 | 0.26 | 0.13 | 0.78 |
| Linear | $T_1$ | 0.51 | 0.53 | 0.09 | 0.42 | 0.63 |
| | $T_2$ | 0.53 | 0.61 | 0.20 | 0.36 | 0.83 |
| LSTM | $T_1$ | 0.33 | 0.35 | 0.09 | 0.24 | 0.48 |
| | $T_2$ | 0.48 | 0.42 | 0.13 | 0.23 | 0.54 |

| **MC-hardness**($\alpha = 0.5$) | | Median | Mean | SD | Min | Max |
|---|---|---|---|---|---|---|
| Layer | Retraining procedure | | | | | |
| S1 Conv 1 | $T_1$ | 0.44 | 0.46 | 0.13 | 0.30 | 0.66 |
| | $T_2$ | 0.63 | 0.69 | 0.18 | 0.56 | 1.00 |
| S2 Conv 1 | $T_1$ | 0.62 | 0.55 | 0.15 | 0.38 | 0.70 |
| | $T_2$ | 0.74 | 0.76 | 0.25 | 0.41 | 1.00 |
| S3 Conv 1 | $T_1$ | 0.61 | 0.59 | 0.19 | 0.34 | 0.81 |
| | $T_2$ | 0.77 | 0.67 | 0.32 | 0.16 | 1.00 |
| Linear | $T_1$ | 0.41 | 0.47 | 0.10 | 0.40 | 0.59 |
| | $T_2$ | 0.57 | 0.65 | 0.22 | 0.45 | 0.92 |
| LSTM | $T_1$ | 0.28 | 0.30 | 0.08 | 0.25 | 0.44 |
| | $T_2$ | 0.49 | 0.43 | 0.16 | 0.24 | 0.62 |

