# OpenReview forum: "Scaling shared model governance via model splitting"
_ICLR.cc/2019/Conference_

### Official Review · AnonReviewer2 · 2018-11-02
**An interesting new nugget of a problem**

**Rating:** 9
**Confidence:** 4

**Review:**

The authors introduce the problem of Model Completion (MC) to the machine learning community.  They provide a thorough review or related works, and convincingly argue that existing solutions to this sort of task (i.e., homomorphic encryption and multi-party computation) are not fully satisfactory in the domain of neural network learning.

The authors also provide extensive numerical experiments attempting to quantify their proposed measure of hardness-of-model-completion, MC-hardness_T(\alpha) on a diverse set of Supervised and RL-related tasks, and they provide extensive analysis of those results.

I find the paper to raise more questions than it answers (in a good way!).  The authors note that their measure depends strongly on the peculiarities of the particular (re)training scheme used.  Do the authors worry that such a measure could end up being too loose--essentially always a function of whatever the fastest optimization scheme happens to be for any particular architecture?

More broadly, there's an additional axis to the optimization problem which is "How much does the training scheme know about the particulars of the problem?", ranging from "Literally has oracle access to the weights of the trained model (i.e., trivial, MC-hardness = 0 always)" to "knows what the architecture of the held-out-layer is and has been designed to optimize that particular network (see, e.g., learned optimizers)" to "knows a little bit about the problem structure, and uses hyperparameter tuned ADAM" to "knows nothing about the problem and picks a random* architecture to use for the held out weights, training it with SGD".

Model completion seems, morally (or at least from a security stand-point) slightly under-specified without being more careful about what information each player in this game has access to.  As it stands, it's an excellent *empirical* measure, and captures a very interesting problem, but I'd like to know how to make it even more theoretically grounded.

An excellent contribution, and I'm excited to see follow-up work.



* We of course have tremendous inductive bias in how we go about designing architectures for neural networks, but hopefully you understand my point.

---

> ### Author Response · Authors · 2018-11-13
> **Yes, there are numerous follow-up directions to explore**
>
> Thank you very much for your thoughtful review!
>
> > Do the authors worry that such a measure could end up being too loose--essentially always a function of whatever the fastest optimization scheme happens to be for any particular architecture?
>
> Yes, this is definitely a concern, and we mention this in the final paragraph. Since we tried a variety of retraining procedures (freezing, not freezing, overparameterization, different initialization schemes) with mostly similar results, we think it is reasonable to be optimistic that there are no radically faster retraining procedures. However, we need to improve our understanding with more experiments, which we have to leave to future work. But we should also stress that at this point the security guarantees this approach provides are only empirical rather than theoretical. Depending on the application that may or may not be a dealbreaker.
>
> > More broadly, there's an additional axis to the optimization problem which is "How much does the training scheme know about the particulars of the problem?"
>
> This is an excellent way to phrase our setup! We decided to strike the balance between impractical retraining procedures like setting all weights to the former values in one step (which requires information that the retraining procedure does not have access to), and making the model completion problem as easy as possible to stress-test its viability as a technique for shared model governance. Since this is (to our knowledge) the first paper on the topic, we struck a balance with what experiments can be executed with reasonable effort; e.g. we did not try to learn a dedicated optimizer. Our goal was to gain some preliminary results on the viability of shared model governance.

---

### Official Review · AnonReviewer1 · 2018-11-03
**Interesting idea but needs better positioning, metrics, and analysis**

**Rating:** 5
**Confidence:** 4

**Review:**

This paper proposes the interesting idea of analyzing how difficult
it is to re-initialize and re-train layers in neural networks.
They study these techniques in the context of ImageNet classification
and reinforcement learning in the Atari and DeepMind lab domains.
While these are interesting ideas and domains to study, I have
concerns with the positioning and execution of the paper.

[Positioning, execution and motivation]
On the positioning of the paper, a significant part of the introduction
and related work section is spent arguing that this approach can be used
for shared model governance in contexts where homomorphic encryption
or secure multi-party computation would instead be used.
Comparing the approaches studied in this paper to these
sophisticated cryptographically-motivated techniques seems
like too much of a stretch, as the methods serve very different
purposes and in most cases cannot even be directly compared.

The first and second paragraph discuss the vision of distributing
the training of models between multiple parties.
I agree that this is a useful area to study and direction
for the community to go, but as the introduction of this paper
states, this is the most interesting when the parties have
control over logically separate components of the modeling pipeline
and also when joint training of the components is being done,
potentially on disjoint and private datasets.
The empirical results of this paper do none of this,
as they only look at the case when a single layer is being
replaced.

Furthermore the motivation and positioning of the paper is
not carried through in the empirical setup, where they
investigate approaches that do training over all
of the parameters of the model, breaking the assumption
that the parties should be independent and should
not share information.

[Metrics for measuring model completeness]
Section 3.1 defines the metric of completion hardness that is
used throughout the rest of the paper. The metric looks at the
number of iterations that re-training the model takes to
reach the same performance as the original model.
It's not clear why this is an important metric and I am
not convinced it is the right one to use as it:
1) does not give a notion of how nicely the missing portion
was recovered, just that the accuracy reached the
same accuracy as the original network, and
2) methods with a very long per-iteration runtime such as
second-order and sampling-based methods could be used to
reach a good performance in a small number of iterations,
making these methods appear to be very "good" at
completing models. I don't think it is nice that this
metric relies on the same optimizer being used for the
original model and the completed model.

I think it's more interesting to study *how much* data is
required to recover missing portions of the model instead
of how many iterations are needed to recover the same performance.
The supervised learning experiments appear to be done
using the entire dataset while the RL experiments do
present a setting where the data is not the same.

[Empirical results]
I am also surprised by the empirical finding in Section 5.1
that T1 outperforms T2, since it seems like only optimizing
the parameters of the missing layer would be the best
approach. I think that if a similarity metric was used
instead, T2 would be significantly better at finding the
layer that is the most similar to the layer that was removed.

Some smaller comments:

1. In Section 3.1, the definition of C_T does not use T explicitly
   inside of it.
2. In the last paragraph of Section 3.1 and first paragraph of
   Section 3.2, N should be defined as an iteration that
   reaches the best loss.
3. The description of T3 does not say what method is used to
   optimize the over-parameterized layer, is it T1 or T2?
4. Why does T4 use T1 instead of T2?
5. In the experimental setup, why is T2 applied with a different
   learning rate schedule than the original training procedure?
6. Why is T2 not shown in the AlexNet results for Figure 2?
7. The dissimilar axes between the plots in Figure 2 and
   Figure 3 make them difficult to compare and interpret.
8. It's surprising that in Figure 3, the hardness of \alpha=1.0
   for T2 is 1.0 for everything.

---

> ### Author Response · Authors · 2018-11-13
> **The model completion problem we study is the hardest realistic test for shared model governance via model splitting**
>
> Thank you very much for the long and thorough review!
>
> > Comparing the approaches studied in this paper to these sophisticated cryptographically-motivated techniques seems like too much of a stretch, as the methods serve very different purposes and in most cases cannot even be directly compared.
>
> These are indeed very different approaches with different overhead/security trade-offs. To our knowledge, the approach we are investigating (model splitting) has not been discussed before and we made some changes in the phrasing to make this more clear in the paper. Both MPC and model splitting try to solve the same problem (shared model governance), but we argue that our approach is more scalable than MPC.
>
> As you point out, a more realistic scenario for model splitting would be one in which the parties do not share the same dataset. However, since we are interested in the security guarantees of model splitting, we study the setting that is hardest to defend against, i.e., the setting that is easiest for the adversary. Since we do not know how much data the adversary has access to, we assume they have access to everything.
>
> > Furthermore the motivation and positioning of the paper is not carried through in the empirical setup, where they investigate approaches that do training over all of the parameters of the model, breaking the assumption that the parties should be independent and should not share information.
>
> We think this could be a misunderstanding. While in retraining procedure T1 we optimize all parameters of the model, there is always at least one entire layer of parameters that has been removed and replaced with a freshly initialized one (See Figure 1 and Section 3.1). This reflects the assumption that there is at least one layer that the adversary would at no point in time have access to, and which is held by the other party. Do you have a sense how we could make this more clear in the paper?
>
> Regarding your points about our metric for the hardness of model completion:
>
> 1. In our experiments we assumed that we only care about the final test accuracy. If there are auxiliary objectives (e.g. being able to fine-tune more easily), these could be added to loss (like a regularizer) and thus feed into the MC-hardness definition. What exactly do you mean by how ‘nicely’ the missing part has been recovered?
>
> 2. This is an excellent point! We wanted to compare computational costs because we are assuming this is what we care about. Computational costs are much harder to compare when you're using different optimizers. This is mentioned in the beginning of section 3.1, where we state our simplifying assumption: *"We assume that computational cost are constant for each step, which is approximately true in our experiments."* We have slightly adjusted the phrasing.
>
> > I think it's more interesting to study *how much* data is required to recover missing portions of the model [...]
>
> This is a very interesting question for the setting in which different parties have different amounts of data. We didn’t study the problem from this angle because we wanted to assume the worst case (the adversary has access to all of the data) in order to stress-test model splitting as an approach for shared model governance. We have to leave the investigation of model completion under partial access to the dataset to future work.
>
> > the RL experiments do present a setting where the data is not the same.
>
> Could you please elaborate how you mean this sentence? The RL experiments train and retrain on the same environment simulator. This simulator is stochastic, so the agent will not be trained on exactly the same data, but on data drawn from the same distribution.
>
> > I am also surprised by the empirical finding in Section 5.1 that T1 outperforms T2.
>
> Your surprise is understandable. However, similar results have also been in the literature; e.g. Figures 1 and 2 in Yosinski et al. (2014) http://papers.nips.cc/paper/5347-how-transferable-are-features-in-deep-neural-networks.pdf.
>
> >  I think that if a similarity metric was used instead, T2 would be significantly better at finding the layer that is the most similar to the layer that was removed.
>
> Could you elaborate how you could use a similarity metric during retraining in our setup? What would you measure similarity to?

---

> > ### Author Response · Authors · 2018-11-13
> > **Regarding your smaller comments**
> >
> > 1. We admit that this is an awkward property of the notation. To make it technically more rigorous, we could add T as a superscript to \theta_i everywhere, but we think this would decrease readability without adding clarity.
> > 2. This is a somewhat arbitrary choice we made. You are right that defining N to be the best loss would also make sense, but the final loss is what we used in our experiments. Moreover, the retraining procedure always starts with the parameters in the final step, not the best parameters. We worry that it would make the formal setup confusing if we used the loss of the best parameters, but used the final parameters for retraining. In practice the difference is minor.
> > 3. It is T1. We have clarified this in the paper.
> > 4. Because T1 has been empirically shown to be a stronger retraining procedure.
> > 5. We tried both and reported the one that performed better. We have clarified this in the paper.
> > 6. Because the results were not very strong. The data is in Appendix C.1, but also added an additional figure (Figure 7) to the appendix.
> > 7. This was a deliberate choice because we think that the comparison between different layers is more meaningful than the comparison between different values of \alpha.
> > 8. In these runs, the model fails to recover the original performance for all layers over the course of the retraining run.

---

> > ### Comment · AnonReviewer1 · 2018-12-03
> > **Updated review**
> >
> > Thanks for the clarifications and for updating some of the motivation
> > parts of the paper. I think this paper has some interesting
> > ideas in it and I have updated my rating from a 4 to a 5
> > after reading the rebuttal. I still echo some of my original
> > concerns that 1) I am not convinced that the number
> > of training iterations to reach some test accuracy is the
> > right metric to use to measure the difficulty of model-completion, and
> > 2) I am not convinced that the experimental tasks demonstrate
> > a useful application or analysis of shared model governance,
> > but I am not an expert in this area.
> >
> > Responding to my smaller comments:
> >
> > > Do you have a sense how we could make this more clear in the paper?
> >
> > On the motivation and positioning, I was referring to the
> > disconnection between parts of the introduction that say
> > "model splitting: distributing a deep learning model between
> > multiple parties such that each party holds a *disjoint* subset
> > of the model’s parameters"
> > and the actual setup that doesn't use a disjoint subset of
> > the parameters (such as T1).
> > Figure 1 also does not capture the T1 re-training procedure
> > as in part (d), layers 1,3,4 appear to stay the same even
> > though T1 could have modified them.
> >
> > > What exactly do you mean by how ‘nicely’ the missing part has been recovered?
> >
> > On the similarity metric for the missing portion, I think
> > it would be interesting to look at the error between the outputs
> > of the original module to the outputs of the re-learned module.

---

> > > ### Author Response · Authors · 2018-12-04
> > > **Questions for further clarification**
> > >
> > > Thank you for your response and for re-evaluating our paper. We appreciate that you are taking the time to engage; the paper has been significantly improved through your feedback.
> > >
> > > Regarding your point 1): What metric do you think we should have used? As stated above, we are interested in the computational cost to reach a certain performance threshold. We could either care about something other than computational costs (such as wall-clock time) or measure performance differently (as you suggested with the similarity metric). While we think both are valid point to modify, but we think that the setting we chose is a reasonable point to start at with this first paper on model completion.
> > >
> > > Regarding your point 2): Could you please elaborate on this point? We wish to know how our experiments could be more convincing or applicable to evaluate model splitting as a technique for shared model governance. The setting we chose is intended to be the most difficult (for the defender) realistic setting.
> > >
> > > Thank you for explaining the similarity metric, that makes a lot of sense. We should look into evaluating this in follow-up work after this paper gets accepted.
> > >
> > > > [...] and the actual setup that doesn't use a disjoint subset of the parameters (such as T1).
> > >
> > > But it does; we think this is where our misunderstanding could lie.
> > >
> > > Let's say the two disjoint subsets of the parameters are (1) the first layer and (2) all the other layers. Suppose we only have access to (2) and not (1) --- i.e., we are the party that holds most, but not all, of the parameters. The re-training procedure T1 then (re)initializes the parameters of the first layer (because we don't have access to them, so we can't use the original weights) and trains all of the parameters jointly: the parameters from (2) and the re-initialized parameters.
> > >
> > > In our experiments we do not hold the disjoint subsets of the parameters on separate machines because our goal is to evaluate the difficulty of model completion. The experimental setup thus artificially removes the part of the model that needs to be completed, pretending that it is unavailable. We made sure that the weights of the missing part do not "leak" into the retraining procedure.
> > >
> > > Does that make sense?
> > >
> > > > Figure 1 also does not capture the T1 re-training procedure as in part (d), layers 1,3,4 appear to stay the same even though T1 could have modified them.
> > >
> > > Yes, thank you for spotting this. We will slightly adjust the colors to reflect that the layers 1,3, and 4 may be finetuned by the retraining procedure.

---

> > > > ### Comment · AnonReviewer1 · 2018-12-04
> > > > **On disjoint subsets**
> > > >
> > > > > > [...] and the actual setup that doesn't use a disjoint subset of the parameters (such as T1).
> > > > >
> > > > > But it does; we think this is where our misunderstanding could lie.
> > > > >
> > > > > Let's say the two disjoint subsets of the parameters are (1) the first layer
> > > > > and (2) all the other layers. Suppose we only have access to (2) and not (1)
> > > >
> > > > I do not think that this is a misunderstanding as I am only trying
> > > > to make a semantic point that the parties don't hold disjoint
> > > > subsets of the parameters, as the introduction to your paper describes.
> > > > In the example you are describing as I see it,
> > > > one party (the originally trained model) will hold all of the
> > > > learned parameters and another party (that is trying to do the completion)
> > > > will hold some subset of the learned parameters and
> > > > some other set of parameters for the part to be completed.
> > > > These two parties are not holding disjoint subsets of the parameters.

---

> > > > > ### Author Response · Authors · 2018-12-04
> > > > > **Re: On disjoint subsets**
> > > > >
> > > > > Apologies for insisting on this point, but we think it's not just a point about semantics. We are worried about a misunderstanding since you stated above that one of your two major critiques of the paper (point 2) is that it is unclear whether "the experimental tasks demonstrate a useful application or analysis of shared model governance."
> > > > >
> > > > > In the example above, the first party would hold *only subset (1) of the parameters* (the first layer, *not* all parameters) and the second party would hold the remaining parameters (2) (layers 2, 3, ...). (See the third paragraph in the introduction.) So both parties hold disjoint subsets of the model, while still being able to train the model jointly by passing activations back and forth. The second party is then the party that tries to do model completion after training using their subset of the weights; in other words, the second party is attacking the security of the shared governance over the trained model.
> > > > >
> > > > > In our experiments, the weights of both parties are on the same machine, and even in the same Tensorflow graph, we are only 'simulating' and attack.
> > > > >
> > > > > Does this clarify the setup?
> > > > >
> > > > > From our discussion, it seems like the paper should make the setup more precise, and we will think about how to improve presentation. Thank you for your help!

---

> > > > > > ### Comment · AnonReviewer1 · 2018-12-04
> > > > > > **Re: On disjoint subsets**
> > > > > >
> > > > > > Thank you for the clarifications on this portion, as there are multiple ways to define the parties for this setup.

---

### Official Review · AnonReviewer3 · 2018-11-03
**Review: Problem motivation and analysis**

**Rating:** 4
**Confidence:** 3

**Review:**

This paper proposes and studies the “model completion” problem: given a trained network (and the data on which is was trained), if a subset of the network is reinitialized from scratch, how many retraining iterations are needed to achieve the original network accuracy (or some percentage of it)? For a variety of networks and problems in both supervised and reinforcement learning, model-completion (MC) hardness is quantified for individual network layers/sections. The experiments are the core of the paper and are generally well documented and seem reproducible.

However, there are two issues that cloud the paper:
	1. The problem motivation (bounding the security of model splitting) is a bit odd. Has model splitting been proposed in the literature as a potential solution to shared model governance? Otherwise it feels like the problem setting was invented to justify the analysis in this paper: “the tail wagging the dog” as the saying goes…
	2. Model completion yet still be an interesting analytical tool for deep networks, but this requires a different evaluation. For instance, model completion provides a way to study how complicated different network layers are to learn or maybe to quantify how much of the inference task may be contained in each. (Though these concepts would need precise language and experimental evidence.) But how do these observations compare to other ways of obtaining similar observations? For instance, from the pruning literature, (Molchanov, 2017, ICLR, https://openreview.net/pdf?id=SJGCiw5gl) includes several figures detailing the statistics of individual network layers and how “prunable" are the filters in each.

This is largely an analytical paper, and I’ll readily acknowledge that it is difficult to pull a clear and insightful study out of a jumble of experimental observations (and hard to review such a paper too). But the limitations of the problem motivation (point #1) and (in my opinion) the misaligned focus of the analysis (point #2), hurt the clarity and significance of this paper. For it to really be a useful tool in understanding deep learning, some additional work seems to be needed.

Other notes:
	3. Pruning literature would be a reasonable comparison in the related work. For instance, (Han, ICLR, 2017, https://arxiv.org/abs/1607.04381) describes a dense-sparse-dense method where a (dense) model is pruned (sparse), after which the pruned connections are reinitialized and retrained (dense) leading to improved accuracy relative to the original dense model.
	4. Consider replacing the uncommonly used “ca.” with “~”, e.g. “~1000x” instead of “ca. 1000x”.
	5. The specifics about ImageNet in the intro to Section 3 should be moved to Section 4.
	6. In Section 3.2 paragraph 2, clarify if “loss” refers to test loss as stated in the intro to Section 3.
	7. In Figure 2 (alpha=0.9) and Figure 3 (alpha=1.0, bottom), why are the values constant?

---

> ### Author Response · Authors · 2018-11-13
> **We made the motivation of our paper more explicit and added related work on pruning**
>
> Thank you very much for your careful review! Let us address your concerns:
>
> 1. We acknowledge that the paper should be more upfront about its motivation (e.g. mention it in the abstract). We are not aware of previous work that discusses model splitting. Our motivation was to find a technique for multi-party computation (MPC) with limited computational & communication overhead. With these constraints, model splitting is an obvious approach to try, and we didn’t think to lay claim to its "invention." We’ve adjusted the phrasing in the paper to make this more clear.
>
> While model splitting is much faster as an MPC technique, its security guarantees are much weaker. The hardness of the model completion problem is at the heart of this MPC technique, and this is the reason we wanted to study it.
>
> 2. Thank you for pointing us to the pruning literature. We have added a paragraph to the related work section.
>
> Regarding the two papers you referenced: Molchanov et al. (2017) evaluate different approaches for pruning trained models with the aim of making inference faster, dropping entire feature maps at a time. Figure 2 in their paper shows the distribution of importance of feature maps across layers (VGG-16). Their findings (e.g. that lower layers are more important) are compatible with ours. There are also other related papers that prune individual connections rather than entire feature maps (e.g. Changpinyo et al., https://arxiv.org/pdf/1702.06257.pdf).
>
> Han et al. (2016) improve the accuracy of models by training in three phases of which the second one adds a sparsity regularizer (how many of the connections are reduced to 0). However, they do not include an analysis on which parts of the model get pruned, making their results incomparable with ours.
>
> All of these papers remove neurons across all layers, i.e. by dropping some but not all neurons in every layer. In contrast, in our experiments we remove entire layers at a time with all of their neurons.
>
> Nevertheless, an interesting connection between our work and the pruning literature you pointed us to is the empirical evidence regarding the relative importance of different layers in the model. However, in contrast to our work the literature does not study what happens when removing an entire layer at a time.
>
> 4-6: Thank you for pointing this out. We have fixed this in the paper.
> 7. In Figure 2 constant values at 0.9 are due to the way learning rate schedule is updated (there's a jump at that point for all layers and they reach 0.9 in the same time). In Figure 3 the values are constant because MC-hardness is capped at 1.0 and none of the layers ever retrieve the full performance.

---

### Author Response · Authors · 2018-11-26
**Updated paper to improve motivation**

To address the reviewers' feedback, we have updated the paper. It should be a lot more clear about the motivation for this work in abstract, introduction, and the paper title. The thank the reviewers for their efforts on in their reviews and hope that their criticism has been addressed appropriately.

---

### Meta-Review · Area_Chair1 · 2018-12-11
**This paper provides some interesting ideas, but has a mismatch between the title and motivation and what is provided**

**Confidence:** 4
**Recommendation:** Reject

**Metareview:**

As all the reviewers have highlighted, there is some interesting analysis in this paper on understanding which models can be easier to complete. The experiments are quite thorough, and seem reproducible. However, the biggest limitation---and the ones that is making it harder for the reviewers to come to a consensus---is the fact that the motivation seems mismatched with the provided approach. There is quite a lot of focus on security, and being robust to an adversary. Model splitting is proposed as a reasonable solution. However, the Model Completion hardness measure proposed is insufficiently justified, both in that its not clear what security guarantees it provides nor is it clear why training time was chosen over other metrics (like number of samples, as mentioned by a reviewer). If this measure had been previously proposed, and the focus of this paper was to provide empirical insight, that might be fine, but that does not appear to be the case. This mismatch is evident also in the writing in the paper. After the introduction, the paper largely reads as understanding how retrainable different architectures are under which problem settings, when replacing an entire layer, with little to no mention of security or privacy.

In summary, this paper has some interesting ideas, but an unclear focus. The proposed strategy should be better justified. Or, maybe even better for the larger ICLR audience, the provided analysis could be motivated for other settings, such as understanding convergence rates or trainability in neural networks.